# The Impact of Hip Mobility on Lumbar and Pelvic Mobility before and after Total Hip Arthroplasty

**DOI:** 10.3390/jcm12010331

**Published:** 2022-12-31

**Authors:** Youngwoo Kim, Claudio Vergari, Yu Shimizu, Hiroyuki Tokuyasu, Mitsuru Takemoto

**Affiliations:** 1Department of Orthopaedic Surgery, Kyoto City Hospital, Kyoto 604-8845, Japan; 2Arts et Métiers Institute of Technology, Institut de Biomécanique Humaine Georges Charpak, Université Sorbonne Paris Nord, 75013 Paris, France; 3International Research Fellow of Japan Society for the Promotion of Science, Nagoya University, Nagoya 464-8601, Japan; 4Department of Rehabilitation, Kyoto City Hospital, Kyoto 604-8845, Japan

**Keywords:** stand-to-sit, sagittal alignment, sagittal balance, lumbar mobility, pelvic mobility, hip mobility, total hip arthroplasty

## Abstract

Hip arthrosis and total hip arthroplasty (THA) can alter a patient’s balance and spinopelvic mobility. In this study, we hypothesized that lumbar, pelvic, and hip mobility and their inter-relations are affected by THA and that their study could give an insight in our understanding of postoperative balance and mobility. **A total of** 165 patients with hip arthrosis and with an indication for THA were included in this single-center prospective cohort. Sagittal radiographs were acquired in four positions: free-standing, standing extension, relaxed-seating and flexed-seating preoperatively and at 6 and 12 months. Spinopelvic parameters were measured (pelvic tilt and incidence, sacral slope, lumbar lordosis, pelvic-femoral angle). Standing spinopelvic parameters did not significantly change postoperatively. However, the postural changes occurring between positions were significantly altered after THA. In particular, pelvic and lumbar mobility was significantly reduced postoperatively, while hip mobility was increased. Correlations were observed between the changes in lumbar, pelvic and hip mobility before and after THA. This study confirmed that there is a relationship between lumbar, pelvic and hip mobility in osteoarthritis, and that this relationship is modified by the postoperative compensation mechanisms deployed by the patient in dynamic postures. Hence, surgeons should consider these relationships when planning surgery, in order to obtain a physiological pelvic tilt postoperatively and to account for the potential increased risk of impingement and dislocation with hip hypermobility.

## 1. Introduction

Stand-to-sit movement is an essential activity in daily living, and it can be divided into two movements: spinopelvic mobility and the hip mobility [1,2,3]. Indeed, the sagittal flexion and extension of the whole kinetic chain is determined by the coordinated motion of the spine, pelvis, and hip [3,4]. From standing to sitting, the pelvis tilts posteriorly by an average of 15° to 20°, and the acetabulum opens approximately 15° to 20° to accommodate the femur, which can flex 55° to 70° [5,6]. This movement can be problematic for patients with hip and spine pathologies [5,7].

Pelvic mobility can be defined as the change in sacral slope (SS) or pelvic tilt (PT) from standing to the sitting position, and it is often categorized as stiff, normal and hypermobile [5]. Several recent studies focused on pelvic mobility because it is directly related to the functional position of the acetabular component after total hip arthroplasty (THA). Similarly, lumbar spinal mobility can be defined as the change in lumbar lordosis (LL) between standing and flexed-seated position [8]. Degenerative change of the lumbar spine induces lumbar stiffness and malalignment, and it decrease spinopelvic mobility during postural changes [2]. Spinal fusion surgery impacts hip-spine biomechanics and also affects the ability to compensate in the stand-to-sit movement [9]. Spinopelvic stiffness and increased femoral motion are associated with late dislocation after THA, because of an increased risk of impingement [10,11].

Hip mobility refers to the motion of the femoroacetabular articulation, and it can be affected by hip pathologies, which can be accompanied by reduced range of motion, hip flexion contracture and severe compensations mechanisms affecting the patient’s alignment [4,12]. Furthermore, restricted hip range of motion can in turn increase lumbar spine impairment and low back pain in patients with hip osteoarthritis (OA).

Recent studies reported that the spinopelvic mobility can change after THA [3,13,14,15]. However, these studies did not assess the full range of motion from standing or extension to flexed-seated position even though the full range of motion of the hip joint is important for spinopelvic mobility. Furthermore, previous study reported that the abnormal spinopelvic characteristics tend to normalize 1 year after THA. Although these studies provide important information regarding the relationship between hip mobility and spinopelvic mobility, no study has evaluated the impact of hip mobility on spinopelvic mobility before and after THA.

The aim of this study was to evaluate the influence of hip mobility on spinopelvic mobility, using radiographic spinopelvic parameters, and their changes from standing (free standing and extension) to sitting (relaxed- and flexed-seated) in a consecutive cohort which was assessed before and 6 and 12 months after THA. Our hypothesis was that spinopelvic mobility would be affected by hip mobility before and after THA.

## 2. Materials and Methods

### 2.1. Patients

This is a single centre, prospective and consecutive cohort of patients. Patients with hip arthrosis and with an indication for THA were included between July 2019 and December 2020 at Kyoto City Hospital (Japan). All procedures were performed by one senior hip surgeons (Y.K.). A modified anterolateral approach was used, wherein the anterior one-fourth of the gluteus medius and minimus muscles were cut at each tendinous portion and the joint capsule was cut in an L-shape. After THA implantation was completed, each gluteus muscles and the joint capsule were sutured using a strong suturing method. A cemented stem and cup were used in all cases. Patients were allowed to begin full weight-bearing and the physical exercise on the second postoperative day. Exclusion criteria were: spinal implant with iliosacral screws, spinal fusion of more than two vertebral levels or scoliosis with coronal Cobb angle higher than 25°. This study was approved by the institutional review board of the Kyoto City Hospital (authorization N. 621) and was conducted per the Helsinki Declaration of 2008.

### 2.2. Data Collection and Radiographic Analysis

Full-body lateral radiographs were acquired in free standing, standing with extension, relaxed-seated and, flexed-seated positions (Figure 1). For the extension radiograph, patients were asked to hold on to a horizontal bar slightly higher than shoulder level, and they were instructed to extend their pelvis and spine as much as possible [4]. The relaxed-seated position is defined as a 90° sitting position, with both femora parallel to the floor on a height- adjustable chair [9]. In the flexed-seated position, the femora are parallel to the floor with the trunk leaning maximally forward [16]. Acquisitions were obtained preoperatively and postoperatively after six and twelve months.

The following standard parameters were measured by an experienced operator in all radiographs: sacral slope (SS), pelvic tilt (PT), pelvic incidence (PI), L1-S1 lumbar lordosis (LL), pelvic incidence minus lumbar lordosis (PI-LL) and pelvic-femur angle (PFA) [11,17].

Spinopelvic and hip mobility were calculated as the change from the standing position (either standing or extension) to a sitting position (either relaxed-seated or flexed-seated), which was indicated as ΔXstanding/sitting = Xsitting-Xstanding. This corresponds to saying, “when sitting, parameter X changed by ΔX degrees”; irrespective of the sign, small values (positive or negative) correspond to small movements.

The dynamic spine-pelvis-hip motion was divided into three mobilities: pelvic, hip and lumbar mobility. Pelvic mobility was defined as the difference in SS between the standing and sitting positions, and it was classified as stiff (ΔSS_standing/relaxed-seated_ ≥ −10°), normal (−10° > ΔSS_standing/relaxed-seated_ > −30°), or hypermobile (ΔSS_standing/relaxed-seated_ ≤ −30°) [5]. Hip mobility was defined as the difference in PFA between the standing and sitting positions (ΔPFA_standing/sitting_). Lumbar mobility was classified according to the change in LL between standing position and flexed-seated position as stiff (ΔLL_standing/flexed-seated_ > −20°), flexible (−20° ≥ ΔLL _standing/flexed-seated_ > −40°) or hypermobile (ΔLL _standing/flexed-seated_ ≤ −40°) [8]. The PI-LL mismatch was measured in the standing position and it was used to assess lumbar spinal balance: a PI-LL > 10° was classified as PI-LL mismatch [18].

### 2.3. Statistics

A preliminary analysis was performed on pilot data, which suggested that a cohort size of 100 patients would allow to detect a postoperative change of PFA of 2° (α = 0.05, β = 0.95) [19]. Differences between preop and postop values were assessed with paired Friedman’s test for multiple comparisons, followed by post hoc Tukey–Kramer analysis. Proportions were compared with z-tests followed by Bonferroni’s correction. Correlations were analysed with Spearman’s rank test. Significance was set at *p* < 0.05 and data was reported as median [quartiles]. Calculations were performed with Matlab 2021b (The Mathworks, Natick, MA, USA).

## 3. Results

One hundred sixty-five patients were included, 137 women and 28 men, median age 70.0 [63.4; 76.0] year old. The median BMI was 23.5 [21.2; 25.8] (Table 1).

### 3.1. Spinopelvic Alignment, Lumbar Mobility and Pelvic Mobility

There was no significant difference in standing LL, SS, PT, PFA, PI, and PI-LL between preoperatively and postoperatively (Table 2). However, several postural changes between standing, extension, relaxed-seated and flexed-seated, were significantly different postoperatively (Table 2). In particular, when moving from standing to the relaxed-seated position, ΔSS was significantly increased (smaller movement) and ΔPT was significantly decreased (smaller movement) 12 months after THA compared with preoperatively (*p* < 0.001 and *p* = 0.002, respectively, Table 3). There was no significant difference in ΔLL from standing to a relaxed-seated position. When moving from a standing to a flexed-seated position, ΔLL and ΔSS was significantly increased (ΔLL: smaller movement, ΔSS: larger movement) and ΔPT was significantly decreased (larger movement) 12 months after THA compared with preoperatively (Table 3).

The number of patients with hypermobile lumbar spine and hypermobile pelvis were both significantly decreased 6 months postoperatively, and it continued to decrease until 12 months (Table 4).

### 3.2. Hip Mobility

When moving from a standing to a relaxed-seated position, ΔPFA was significantly increased 12 months after THA compared with preoperatively (ΔPFA: 63.8° compared with 59.4°; *p* < 0.001, Table 3). Similarly, ΔPFA when moving from standing to a flexed-seated position significantly increased postoperativelyΔPFA from standing to a relaxed-seated position or a flexed-seated position in stiff pelvis group (ΔSS ≥ −10°) was significantly larger than normal (−10° > ΔSS > −30°) and hypermobile pelvis group (ΔSS ≤ −30°, Figure 2). There was no significant difference between ΔPFA before and after THA in the stiff group. However, ΔPFA from standing to a relaxed- or flexed-seated positions in normal and hypermobile group were significantly increased 6 months and 12 months after THA compared with preoperatively.

### 3.3. Relationship between Pelvic Mobility, Lumbar Mobility, and Hip Mobility before and after THA

From the standing to the relaxed-seated position, the change of ΔPFA between preoperatively to 12 months postoperatively was correlated with the change of ΔSS (R = 0.3, *p* < 0.01, Figure 3). Interestingly, the change of ΔPFA from a standing or extension to a flexed-seated position was strongly correlated with the change of ΔSS (r = 0.8, *p* < 0.001). The change of ΔSS from a standing to a relaxed-seated position between preoperatively and 12 months postoperatively was also strongly correlated with the change of ΔLL (R = −0.8, *p* < 0.001) (Figure 4). Overall sagittal flexion (ΔLL + ΔPFA) from a standing or extension to a flexed-seated position significantly increased 6 month and 12 months after THA in patients with hypermobile pelvis, compared with preoperatively (Figure 5), but not in normal and stiff pelvis.

Postoperative PFA in relaxed-seated position in patients with preoperative hypermobile pelvis (48.3° [37.4, 58.6]) was lower than in patients with normal pelvis (59.3° [48.8, 68.6], *p* = 0.02) and stiff pelvis (57.7° [48.7, 69.0], *p* = 0.02).

## 4. Discussion

This study gives an insight into spinopelvic and hip mobility in osteoarthritis, what is their mutual relationship and how they change between different postures and postoperatively. Care was taken to define lumbar, pelvic and hip mobility, since these terms are sometimes used loosely in the literature [5,20]. Results show that hip mobility (ΔPFA) in four positions (free-standing, extension, relaxed-seated and flexed-seated.) was correlated with the change of pelvic mobility (ΔSS) before and one year after primary THA. It was also found that pelvic and lumbar mobility (ΔSS and ΔLL) was significantly decreased one year after THA, while hip mobility (ΔPFA) was significantly increased. These results support this study’s hypothesis that hip mobility influences spinopelvic mobility, since the mobility of the pelvis is driven by the mobility of the hip joint. The factors affecting spinopelvic mobility are of increasing interest in patients after THA [1,5,21,22], and it is important for arthroplasty surgeon to understand the complex relationship between the hip joint, pelvis and the lumbar spine in order to identify high risk patients for dislocation and impingement after THA. However, these factors are still unknown. The present study identified the importance of the hip mobility in determining pelvic tilt in standing and sitting position before and after THA.

Spinopelvic mobility plays an important role in functional acetabular component position following THA. Physiologically, the pelvis rotates in retroversion from standing to sitting position, but this movement can be influenced by spinopelvic and hip mobility. Decreased mobility in pelvis retroversion can cause functional cup retroversion and increases the risk of anterior impingement and posterior dislocation [1,5,23]. Previous studies reported that spinopelvic stiffness can be due to a stiff spine, such as in spinal fusion surgeries and biological spinal fusion [2,9,21]. However, our study indicated that lumbar and pelvic mobility from standing to relaxed-seated position was decreased 1 year after THA, and significantly less patients showed hypermobile pelvis and lumbar spine after surgery, while more patients showed stiff spines, although there was no significant difference in the standing PI-LL nor in the distribution of PI-LL mismatch between preoperatively and postoperatively. These results indicated that the decrease in the pelvic and lumbar mobility may not be due to a degeneration of the lumbosacral region, but rather by an improved the range of motion of the hip joint after THA. In other words, THA restored some hip range of motion, which allowed the patients to use this joint in the flexed position, rather than flexing their lumbar spine. Previous studies supported this interpretation of the results, in the sense that restricted hip mobility influences spinopelvic mobility [1,24,25].

Hip pathology can be accompanied by hip flexion contracture and reduced range of motion of the hip in standing and sitting position [4]. Recent studies reported that hip pathology can lead to posterior pelvic tilt in the sitting position because of the contracture, and this movement of the pelvis is associated with a compensatory increased lumbar flexion [16]. This spinopelvic hypermobility was resolved one year after THA [3,25]. Our results also confirmed that ΔSS from a standing to a relaxed-seated position was significantly decreased and ΔLL from a standing to a flexed-seated significantly decreased one year after THA. One reason for this postoperative change is thought to be due to resolution of preoperative hip contracture [25].

Increased pelvis mobility in anteversion after THA can be risky. The anterior movement of the pelvis from standing to a relaxed- and flexed-seated position cause functional cup retroversion and increases the risk of anterior impingement [21]. Previous study reported that this adverse anterior pelvic tilt (≥20°) from standing to flexed-seated position is a risk factor for dislocation after THA [21]. The present study described that there is strong correlation between the change of ΔPFA and ΔSS between preoperative and postoperative from standing to flexed-seated position. Hence, hip mobility can be considered one of the factors determining the anterior pelvic movement from standing to flexed-seated position. Improvement of the range of motion of hip joint after THA might be a risk factor for dislocation after THA. Hypermobility in the anterior direction should be assessed using radiographs in the lateral flexed-seated position. This position was also useful because it is associated with maximal sagittal flexion of the kinematic chain. Our results indicated the overall sagittal flexion (ΔLL + ΔPFA) from standing or extension to relaxed- and flexed-seated position was increased after THA in patients with hypermobile pelvis. The flexed-seated position is a risk for anterior femoroacetabular impingement, and therefore this position can better highlight spinal compensatory mechanism in radiological examination.

Patients with spinopelvic stiffness from standing to sitting position are at high risk of hip dislocation after THA [10]. Spinopelvic stiffness is a well-established parameter which can be measured using dynamic standing and sitting lateral radiographs [1]. This study demonstrated that the change of ΔSS was correlated with ΔPFA from standing to relaxed- and flexed-seated position between preoperative and postoperative. Thus, the postoperative lumbar and pelvic mobility was decreased without spinal degeneration and the improvement of hip mobility affected this lumbar and pelvic mobility. A recent study supports our result that ΔSS_standing/relaxed-seated_ ≥ −10° was not correlated with a stiff spine and overpredicted the presence of stiff spine [26]. Our results suggest that spinopelvic assessment using only ΔSS might be not sufficient, and hip mobility in the relaxed- and flexed-seated position should also be assessed before THA. Furthermore, the modification of the preoperative planning of cup anteversion and abduction angle for THA might be required depend on hip mobility preoperatively. However, postoperative PFA is difficult to predict preoperatively because postoperative PFA can depend on the surgical technique, limb lengthening and postoperative care. The prediction of the postoperative PFA should be explored in future studies.

This study has several limitations. First, only a small number of subjects were investigated due to the strict inclusion and exclusion criteria. Despite having enough statistical power, these finding need to be confirmed in a larger patient population. Second, we have excluded patients with previous spinal fusion of more than two levels. The finding of this study should also be investigated among those patients as their spinopelvic mobility change after THA might be different. These limitations should be considered when interpreting the results and should be addressed in future studies.

## 5. Conclusions

This study confirmed that the pelvic mobility (ΔSS), lumbar mobility (ΔLL) and hip mobility (ΔPFA) are inter-related before and after THA. Hip mobility was improved after THA and the change of hip mobility between preoperative and 12 months postoperative was correlated with the change of pelvic mobility from standing to relaxed- and flexed-seated position. The results suggest that hip mobility is one of the factors determining pelvic tilt before and after THA. Future studies must investigate how those preoperative and postoperative changes in hip, lumbar and pelvic mobility lead to cup position outside of their normal safe range. However, at stage, surgeons should already take into account that patients with preoperative hypermobile pelvis could be at risk of postoperative posterior impingement, because of low PFA, while hypermobility in hip flexion can increase risk of anterior impingement.

## Figures and Tables

**Figure 1 jcm-12-00331-f001:**
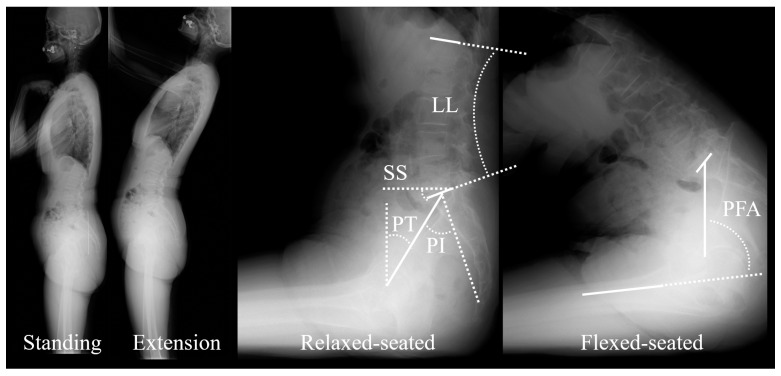
Lateral radiographs in free standing (Standing), standing with extension (Extension), relaxed-seated and, flexed-seated positions. Main radiological parameters are reported: sacral slope (SS), pelvic tilt (PT), pelvic incidence (PI), lumbar lordosis (LL), and pelvic-femur angle (PFA).

**Figure 2 jcm-12-00331-f002:**
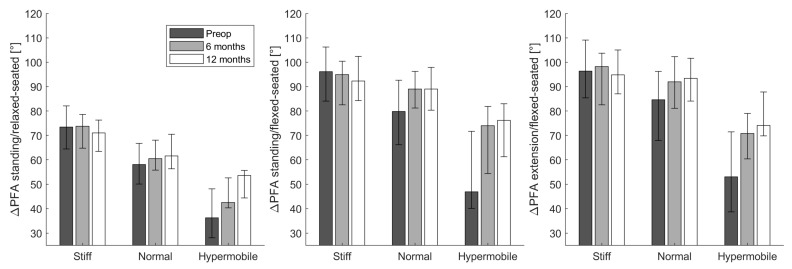
The change of pelvic-femoral angle (ΔPFA) from the standing or extension to the relaxed-or flexed- seated position in stiff (ΔSS ≥ −10°), normal (−10° > ΔSS > −30°), or hypermobile (ΔSS ≤ −30°) groups at three different stages (preoperatively (preop), 6 months and 12 months after THA).

**Figure 3 jcm-12-00331-f003:**
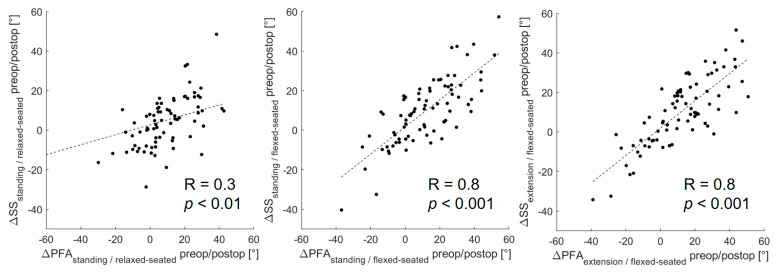
Correlation between the change of ΔPFA and ΔSS between preoperative and 12 months postoperative form standing or extension to relaxed- or flexed-seated position.

**Figure 4 jcm-12-00331-f004:**
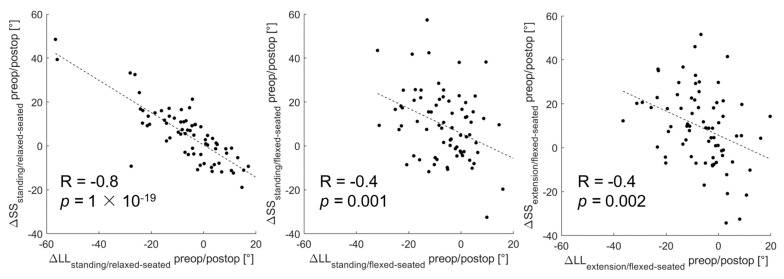
Correlation between the change of ΔLL and ΔSS between preoperative and 12 months postoperative form standing or extension to relaxed- or flexed-seated position.

**Figure 5 jcm-12-00331-f005:**
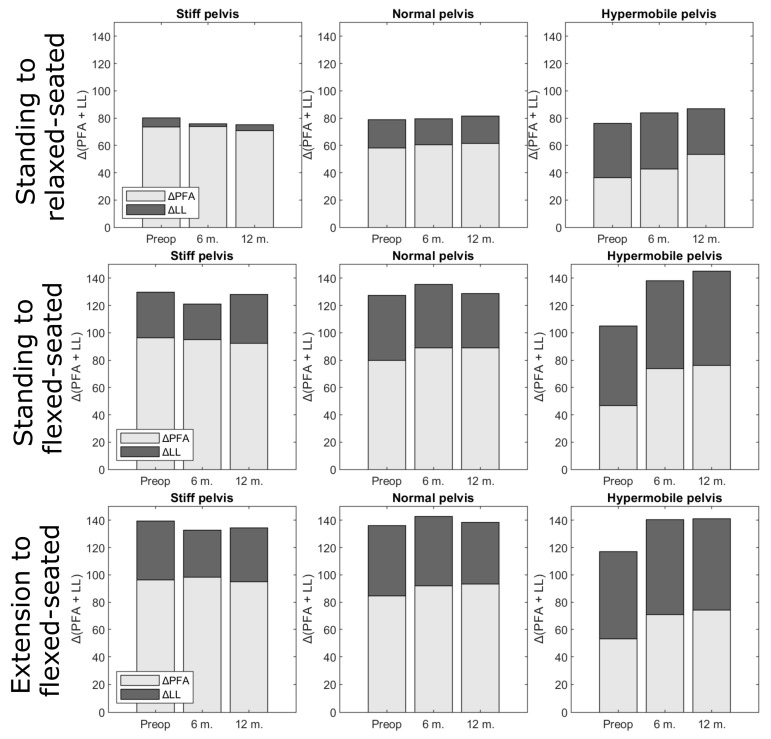
Overall sagittal flexion (ΔLL + ΔPFA) from standing or extension to relaxed- or flexed-seated position, at three different stages (preoperatively (preop), 6 months (6 m) and 12 months (12 m) after THA), according to pelvis mobility. Proportions were not significantly different.

**Table 1 jcm-12-00331-t001:** Demographics of the cohort.

	Cohort
**Number**	165
**Female/male**	137/28
**Age**	70.0 [63.4; 76.0]
**Body mass index**	23.5 [21.2; 25.8]
**Pelvic incidence**	46.3 [38.3; 54.8]

**Table 2 jcm-12-00331-t002:** Spinopelvic parameters of the patients in standing, extension, relaxed-seated, and flexed-seated positions preoperatively (preop.) and 6 months (6 m) and 12 months (12 m) postoperatively (postop.). Values are reported as median [quartiles], and *p*-values are reported for significant differences between preoperatively and postoperatively. Parameters are: L1-S1 lumbar lordosis (LL), sacral slope (SS), pelvic tilt (PT), pelvic-femur angle (PFA), pelvic incidence (PI), pelvic incidence minus lumbar lordosis (PI-LL), and PI-LL.

	Position	Preop.	6 m Postop.	12 m Postop.	*p* (Preop. vs. 6 m)	*p* (Preop. vs. 12 m)	*p* (6 m vs. 12 m)
**LL [°]**	Standing	42 [30; 52]	42 [29; 54]	43 [31; 54]	0.662	0.974	0.793
	Extension	48 [37; 57]	47 [35; 56]	48 [37; 57]	0.021	0.021	*p* < 0.001
	Relaxed-seated	20 [7; 34]	24 [12; 37]	25 [14; 36]	0.039	0.032	0.997
	Flexed-seated	−5 [−14; 6]	−3 [−15; 8]	−1 [−11; 9]	0.294	*p* < 0.001	0.039
**SS [°]**	Standing	32 [25; 41]	33 [25; 40]	33 [25; 41]	0.491	0.714	0.933
	Extension	28 [20; 36]	26 [19; 34]	28 [19; 35]	0.221	0.356	0.955
	Relaxed-seated	15 [6; 24]	16 [8; 24]	17 [11; 25]	0.148	0.022	0.709
	Flexed-seated	39 [26; 51]	47 [33; 5]	49 [37; 59]	0.014	*p* < 0.001	*p* < 0.001
**PT [°]**	Standing	14 [9; 21]	15 [10; 22]	15 [10; 20]	0.753	0.441	0.870
	Extension	19 [12; 26]	19 [14; 27]	20 [16; 26]	0.997	0.473	0.430
	Relaxed-seated	32 [24; 44]	31 [23; 40]	31 [24; 37]	0.214	0.001	0.140
	Flexed-seated	10 [−2; 26]	5 [−5; 15]	2 [−7; 13]	0.009	*p* < 0.001	*p* < 0.001
**PFA [°]**	Standing	−3 [−12; 4]	−5 [−13; 0]	−5 [−11; 0]	0.032	0.067	0.956
	Extension	−8 [−15; −1]	−9 [−16; −2]	−9 [−14; −3]	0.590	0.122	0.590
	Relaxed-seated	57 [45; 66]	57 [48; 67]	59 [52; 66]	0.177	0.011	0.507
	Flexed-seated	80 [61; 91]	83 [70; 92]	85 [75; 94]	0.047	*p* < 0.001	*p* < 0.001
**PI [°]**	Standing	46 [38; 55]	46 [39; 56]	46 [39; 57]	0.224	0.753	0.619
**PI-LL [°]**	Standing	6 [−2; 16]	7 [−2; 19]	7 [−2; 18]	0.997	0.348	0.311
	Extension	0 [−8; 11]	2 [−7; 10]	1 [−7; 11]	0.546	0.328	0.926
	Relaxed-seated	29 [16; 43]	24 [16; 36]	25 [16; 34]	0.014	0.004	0.915
	Flexed-seated	55 [47; 63]	54 [44; 61]	52 [43; 61]	0.016	*p* < 0.001	0.167
**PI-LL mismatch (%)**	Standing	38	44	38			
	Extension	26	24	28			
	Relaxed-seated	89	85	87			
	Flexed-seated	99	100	100			

**Table 3 jcm-12-00331-t003:** The changes in the spinopelvic parameters of the patients between standing, extension, relaxed-seated, and flexed-seated positions preoperatively (preop.) and 6 (6 m) and 12 months (12 m) postoperatively (postop.). Values are reported as median [quartiles], and *p*-values are reported for significant differences between preoperatively and postoperatively. Parameters are: L1-S1 lumbar lordosis (LL), sacral slope (SS), pelvic tilt (PT), and pelvic-femur angle (PFA).

	Position	Preop.	6 m Postop.	12 m Postop.	*p* (Preop. vs. 6 m)	*p* (Preop. vs. 12 m)	*p* (6 m vs. 12 m)
**ΔLL [°]**	Standing to relaxed-seated	−19 [−32; −9]	−15 [−25; −5]	−16 [−26; −6]	0.102	0.145	0.985
	Standing to flexed-seated	−46 [−56; −33]	−44 [−54; −29]	−39 [−52; −32]	0.645	*p* < 0.001	0.006
	Extension to flexed-seated	−51 [−63; −40]	−47 [−58; −35]	−44 [−59; −35]	0.028	*p* < 0.001	0.258
**ΔSS [°]**	Standing to relaxed-seated	−17 [−25; −9]	−14 [−22; −7]	−14 [−21; −6]	0.050	*p* < 0.001	0.426
	Standing to flexed-seated	7 [−7; 20]	13 [2; 24]	17 [6; 27]	0.053	*p* < 0.001	*p* < 0.001
	Extension to flexed-seated	12 [−2; 23]	18 [10; 27]	21 [10; 32]	0.012	*p* < 0.001	0.003
**ΔPT [°]**	Standing to relaxed-seated	18 [10; 28]	16 [9; 23]	14 [9; 21]	0.364	0.002	0.104
	Standing to flexed-seated	−5 [−17; 10]	−11 [−21; −2]	−13 [−24; −4]	0.146	*p* < 0.001	*p* < 0.001
	Extension to flexed-seated	−8 [−22; 5]	−14 [−24; −4]	−17 [−26; −7]	0.122	*p* < 0.001	0.001
**ΔPFA [°]**	Standing to relaxed-seated	59 [48; 72]	63 [56; 72]	64 [58; 73]	0.137	*p* < 0.001	0.083
	Standing to flexed-seated	82 [64; 96]	89 [78; 97]	91 [81; 98]	0.122	*p* < 0.001	0.003
	Extension to flexed-seated	86 [67; 99]	91 [80; 101]	94 [84; 103]	0.059	*p* < 0.001	0.059

**Table 4 jcm-12-00331-t004:** The distribution of the patients preoperatively (preop.) and 6 (6 m) and 12 months (12 m) postoperatively (postop.) for pelvic mobility (Stiff, normal and hypermobile (hyper)), lumbar mobility (Stiff, flexible, and hypermobile).

Mobile Type	Preop.	6 m Postop.	12 m Postop.	Differences between Stages
**Pelvic mobility**				
Stiff (ΔSS ≥ −10°)	29%	30%	36%	
Normal (−10° > ΔSS > −30°)	55%	61%	60%	
Hypermobile (ΔSS ≤ −30°)	16%	9%	4%	Preop. vs. 6 m; Preop. vs. 12 m
Differences between groups	Stiff vs. Hyper; Normal vs. Hyper	Normal vs. Hyper	Normal vs. Hyper	
**Lumbar mobility**				
Stiff (ΔLL > −20°)	6%	10%	8%	
Flexible (−20° ≥ ΔLL ≥ −40°)	32%	31%	47%	
Hypermobile (ΔLL < −40°)	62%	59%	45%	Preop. vs. 12 m; 6 m vs. 12 m
Differences between groups	Stiff vs. Flexible; Stiff vs. Hyper	Stiff vs. Flexible; Stiff vs. Hyper	Stiff vs. Flexible; Stiff vs. Hyper; Flexible vs. Hyper	

ΔSS (Standing to relaxed-seated); *p* < 0.001, ΔLL (Standing to flexed-seated); *p* < 0.001.

## Data Availability

Not applicable.

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
