# Peer review of "The Impact of Hip Mobility on Lumbar and Pelvic Mobility before and after Total Hip Arthroplasty"

_jcm, 2022, doi:10.3390/jcm12010331_

Round 1
Reviewer 1 Report
The scientific work appears to be correctly set up and carried out but the topic, although interesting in its practical implications on hip prosthetic surgery, does not add significant elements to the notions already acquired, reiterating the functional spino pelvic relationship with the hip joint. The authors themselves emphasize how the influence of hip prosthesization on this relationship, is conditioned by surgical technique and rehabilitation, but without giving any new elements and knowledge in this regard to guide pratically surgeons who routinely perform this type of surgery. Given the complexity of the topic, it would be desirable to collect more data also related to the surgical technique, the performance or not of capsular release, and the rehabilitation course to clarify how the surgical technique used in hip prosthesis affects spino pelvic mobility
Author Response
Instructions for Documenting Changes Done in Revisions
Submitted to Journal of Clinical Medicine
The authors are grateful to the reviewers for their comments which allowed us to improve the paper and clarify its objective. Please find below specific answers.
|
Numbered Reviewer Remark and Manuscript Line Number |
Author Response |
Revised Manuscript Line Number and Text Change |
|
Reviewer 1:
|
|
|
|
1) The scientific work appears to be correctly set up and carried out but the topic, although interesting in its practical implications on hip prosthetic surgery, does not add significant elements to the notions already acquired, reiterating the functional spino pelvic relationship with the hip joint. |
Thank you for your comments. The strengths of our study included a detailed assessment of mobility in all three regions both preoperatively and postoperatively to determine their relationship to one another. We think it is unique compared to prior literature on the subject. More clinical application of the assessment of hip mobility was added. However, we agree that the practical implications on surgical strategy was not sufficiently emphasized. This was amended in the current version, as we added practical suggestions to implement in surgical planning.
|
Line 243: Furthermore, the modification of the preoperative planning of cup anteversion and abduction angle for THA might be required depend on hip mobility preoperatively.
Line 265: However, at stage, surgeons should already take into account that patients with preoperative hypermobile pelvis could be at risk of postoperative posterior impingement, because of low PFA, while hypermobility in hip flexion can increase risk of anterior impingement.
|
|
The authors themselves emphasize how the influence of hip prosthesization on this relationship, is conditioned by surgical technique and rehabilitation, but without giving any new elements and knowledge in this regard to guide pratically surgeons who routinely perform this type of surgery. Given the complexity of the topic, it would be desirable to collect more data also related to the surgical technique, the performance or not of capsular release, and the rehabilitation course to clarify how the surgical technique used in hip prosthesis affects spino pelvic mobility |
We agree with the reviewer in that surgical technique, the performance or not of capsular release, and the rehabilitation are important aspects. More information about these were added. |
Line 66: A modified anterolateral approach was used, wherein the anterior one-fourth of the gluteus medius and minimus muscles were cut at each tendinous portion and the joint capsule was cut in an L-shape. After THA implantation was completed, each gluteus muscles and the joint capsule were sutured using a strong suturing method.
Line 70: Patients were allowed to begin full weight-bearing and the physical exercise on the second postoperative day. |
Reviewer 2 Report
First of all I would like to congratulate the authors who have designed and carried out this very interesting study
In text lines 110-118 the > and < symbols seem to be used improperly, probably should be reversed also accordingly with the cited article.
also it is not clear to me why ΔSS, the difference between standing SS and sitting SS is expressed with negative numbers but the calculated difference is a positive number (standing SS – relaxed sitting SS = 32 -15 =17); same goes for the ΔLL. The result is actually not a negative number, and expressed in this way, I personally find the data rather confusing.
I found no mention of the term flatback in the cited article by Frank J. Schwab, MD (text line 118)
I do not have sufficient knowledge of statistics to judge the proposed analysis, which in any case appears to be described in sufficient detail to be criticized by experts.
The sample size is more than adequate considering the preliminary analysis conducted on pilot data.
The results are presented comprehensively (although, as far as I am concerned, I continue to find the use of > < symbols and negative numbers for actual positive values confused, I don't know if it's my limit also linked to my not being a native English speaker ) the tables and graphs are correctly presented and the data are very interesting and underline the interdependence between hip mobility (also prosthetic) and mobility of the lumbar spine and pelvis.
The discussion and conclusions are consistent with what is described in the materials methods and results and the information provided offers many food for thought for both vertebral surgeons and hip surgeons.
The close functional correlation between hip and lumbar spine is, in some ways well known but for others it continues to be elusive to our understanding, especially in terms of clinical utility. There are still several elements to translate into practice, therefore in a precise orientation of the hip prosthetic components that takes into account the spinopelvic biomechanics.
This article has the merit not only of keeping alive the attention to this topic but also of providing new interesting data.
In my opinion, however, the problem of symbols <> and positive values expressed with negative numbers needs to be clarified.
Author Response
Instructions for Documenting Changes Done in Revisions
Submitted to Journal of Clinical Medicine
The authors are grateful to the reviewers for their comments which allowed us to improve the paper and clarify its objective. Please find below specific answers.
|
Numbered Reviewer Remark and Manuscript Line Number |
Author Response |
Revised Manuscript Line Number and Text Change |
|
Reviewer 2:
|
|
|
|
First of all I would like to congratulate the authors who have designed and carried out this very interesting study In text lines 110-118 the > and < symbols seem to be used improperly, probably should be reversed also accordingly with the cited article. |
Thank you for your comments. The cited article uses the opposite approach: “when standing up, SS increases by X degrees”. This choice was explained in the present revision.
|
Line 91: Spinopelvic and hip mobility were calculated as the change from the standing position (either standing or extension) to a sitting position. Line 93: This corresponds to saying, “when sitting, parameter X changed by ΔX degrees”; irrespective of the sign, small values (positive or negative) correspond to small movements. |
|
also it is not clear to me why ΔSS, the difference between standing SS and sitting SS is expressed with negative numbers but the calculated difference is a positive number (standing SS – relaxed sitting SS = 32 -15 =17); same goes for the ΔLL. The result is actually not a negative number, and expressed in this way, I personally find the data rather confusing.
|
SS is actually expressed as positive values (see Table 2), while the difference can indeed be negative since SS decreases when sitting down. So in the example made by the reviewer, ΔSS = SSsitting – SSstanding = 15° – 32° = -17°. The same applies to ΔLL = 20°-42° = -19°. The principle is the same as with opposite values: small values (positive or negative) correspond to small movements, or “stiffness”. This was also clarified (see answer above) |
|
|
I found no mention of the term flatback in the cited article by Frank J. Schwab, MD (text line 118) |
The definition was corrected. |
Line 102: a PI-LL > 10° was classified as PI-LL mismatch [20]. |
|
I do not have sufficient knowledge of statistics to judge the proposed analysis, which in any case appears to be described in sufficient detail to be criticized by experts. The sample size is more than adequate considering the preliminary analysis conducted on pilot data. The results are presented comprehensively (although, as far as I am concerned, I continue to find the use of > < symbols and negative numbers for actual positive values confused, I don't know if it's my limit also linked to my not being a native English speaker ) . the tables and graphs are correctly presented and the data are very interesting and underline the interdependence between hip mobility (also prosthetic) and mobility of the lumbar spine and pelvis. The discussion and conclusions are consistent with what is described in the materials methods and results and the information provided offers many food for thought for both vertebral surgeons and hip surgeons. |
Thank you for your comments, which made us realize the potentianl confusion concerning the use of the “sitting-standing” convention. Hopefully the confusion should be clarified with the latest modifications. |
|
|
The close functional correlation between hip and lumbar spine is, in some ways well known but for others it continues to be elusive to our understanding, especially in terms of clinical utility. There are still several elements to translate into practice, therefore in a precise orientation of the hip prosthetic components that takes into account the spinopelvic biomechanics. |
Our study is a stepping stone towards a full understanding of the role of the hip in full sagittal balance, but it already allows to provide practical advice for surgeons; this was made explicit in the revised conclusion. |
End of conclusion: However, at stage, surgeons should already take into account that patients with preoperative hypermobile pelvis could be at risk of postoperative posterior impingement, because of low PFA, while hypermobility in hip flexion can increase risk of anterior impingement. |
|
This article has the merit not only of keeping alive the attention to this topic but also of providing new interesting data. In my opinion, however, the problem of symbols <> and positive values expressed with negative numbers needs to be clarified.
|
Thank you for your comment. |
|